# Farmed oyster mortality follows consistent Vibrio community reorganization

Steph Smith,[1] Mark Ciesielski,[1] Thomas Clerkin,[1] Tal Ben-Horin,[2] Rachel T. Noble[1]

**ABSTRACT** Mortality events in marine bivalves cause substantial economic losses in aquaculture, yet the microbial dynamics underlying these events remain poorly characterized. Here, we investigated succession patterns in oyster-associated Vibrio communities during mortality events by sampling eastern oysters (*Crassostrea virginica*) at a North Carolina commercial farm that has experienced repeated, unexplained mortality events. Through whole-genome sequencing of 110 Vibrio isolates from 26 oysters sampled across mortality events in two consecutive years, we identified six conserved phylogenetic clades with distinct temporal associations. *Vibrio mediterranei* and a clade of resident vibrios consistently dominated the initial cultured community at the onset of mortality. However, *V. mediterranei* was absent as mortality progressed, coinciding with increased abundance of *V. harveyi*, *V. alginolyticus*, *V. diabolicus*, and *V. agarivorans*. Comparative genomic analysis revealed that initial isolates were enriched in pathways associated with host colonization and complex carbon metabolism, while isolates from elevated mortality showed enrichment in virulence mechanisms and adaptation to degraded host tissues. Temporal separation between genetically distinct clades suggests microbial competition shapes community assembly during mortality events that ultimately reached >85% mortality in both years. This predictable succession from commensal to potentially pathogenic Vibrio species provides genome-level insight into microbial community dynamics during oyster mortality. The consistent loss of *V. mediterranei* prior to severe mortality suggests this species could serve as a bioindicator for early warning systems to mitigate economic losses in shellfish aquaculture.

**IMPORTANCE** Mortality events in aquaculture systems represent complex host–microbe-environment interactions that challenge our ability to predict and prevent disease. By characterizing succession patterns in oyster-associated Vibrio communities at whole-genome resolution, we reveal a consistent transition from metabolically versatile species associated with healthy oysters to functionally distinct Vibrio taxa enriched in virulence factors and tissue degradation pathways as mortality progresses. This genome-level evidence for predictable community reorganization suggests that monitoring commensal Vibrio populations, particularly the presence or absence of *Vibrio mediterranei*, could provide earlier warning of impending disease compared to tracking only known pathogens. This shift in monitoring approach could advance aquaculture disease management while expanding our fundamental understanding of how microbial community transitions contribute to host health and disease progression.

**KEYWORDS** *Vibrio*, microbial succession, ecological succession, oyster microbiome, shellfish aquaculture, mortality dynamics, comparative genomics, host–microbe interactions, early warning indicators

Mortality events in marine bivalves represent a significant threat to global aquaculture production, causing substantial crop losses and threatening food security.

**Peer Reviewer** Jeffrey W. Turner, Texas A&M University at Corpus Christi, Corpus Christi, Texas, USA

Address correspondence to Steph Smith, steph.smith@unc.edu.

The authors declare no conflict of interest..

See the funding table on p. 15.

These events impact all major shellfish aquaculture species, including oysters, clams, and mussels, with some outbreaks resulting in greater than 90% stock losses in affected areas (1). While these events have been documented for decades (2–4), our understanding of the underlying disease dynamics, mechanisms, and etiology remains incomplete. This is particularly challenging in eastern oysters (*Crassostrea virginica*), where discrete mortality events vary in duration, severity, and physiological manifestation, complicating efforts to determine disease etiology and develop effective management strategies.

Recent pathological investigations of eastern oyster mortality events in North Carolina, USA, have identified the digestive system as a primary site of tissue damage consistent with potential bacterial infection (5). Vibrios are often implicated as drivers of mortality events across marine organisms (1, 6, 7), yet their ecological roles can span from beneficial symbionts to opportunistic pathogens depending on environmental and host context. For example, acquisition of a virulence plasmid resulted in the evolution of *Vibrio crassostreae* from a benign oyster commensal to a documented oyster pathogen (8).

A consistent feature of mortality events across bivalve species is dysbiosis, a disruption of the host-associated microbial community that either precedes or coincides with mortality. In Pacific oysters infected with OsHV-1 microvariants, dysbiosis serves as an intermediate step toward mortality, where the loss of beneficial bacteria weakens oyster resilience while opportunistic Vibrios proliferate (3). Microbiome dysbiosis has been repeatedly documented during shellfish mortality events (9–11), but whether these changes drive host decline or merely respond to altered host physiology remains unclear. This fundamental question has significant implications for guiding intervention strategies in aquaculture systems.

Environmental parameters, particularly temperature and salinity, influence Vibrio community composition and virulence expression (12–15). Vibrio species associated with mortality events often harbor conserved virulence factors that are differentially regulated by environmental conditions and host factors. These virulence factors include type III secretion systems (T3SS), hemolysins, and metalloproteases that directly impact host tissues (1, 16). While environmental fluctuations likely contribute to the seasonal nature of mortality events, the specific mechanisms linking environmental changes to Vibrio community succession remain poorly characterized in eastern oysters, particularly regarding the temporal dynamics of community transitions.

In this study, we characterized Vibrio community dynamics during oyster mortality events at genome-level resolution by sequencing 110 isolates from oyster visceral tissues across discrete mortality thresholds. Over 2 years, we tracked eastern oysters deployed at a commercial aquaculture lease in North Carolina that has experienced repeated mortality events in the past decade. By examining Vibrio dynamics throughout multiple mortality events at a single farm, we investigate how Vibrio populations reorganize as mortality events progress while controlling for geographic variation.

Our findings reveal consistent patterns of community reorganization with distinct Vibrio species associated with healthy and diseased population states. At the onset of mortality, communities are dominated by *Vibrio mediterranei*, which possess unique metabolic capabilities that suggest potential beneficial or neutral interactions with the host. As mortality progresses, the Vibrio community transitions to dominance by *V. harveyi*, *V. alginolyticus*, *V. diabolicus*, and *V. agarivorans*—species enriched in stress response mechanisms and virulence factors. By integrating community dynamics with comparative genomics, this study provides new insights into how Vibrio-oyster interactions evolve during mortality events. Understanding these patterns may help identify early warning signs of impending mortality and suggest potential intervention points for aquaculture management.

## MATERIALS AND METHODS

### Study site and sample collection

Eastern oysters (*Crassostrea virginica*) were deployed at a commercial aquaculture lease in North Carolina that experiences annual mortality events. Diploid (2N) and triploid (3N) oysters from the HNRY breeding line were deployed in April of each year (2022–2023) and monitored between April and May. Environmental parameters (water temperature, salinity, dissolved oxygen) were collected at each sampling event with a YSI-EXO2 multi-parameter water quality sonde (YSI Inc./Xylem Inc., Yellow Springs, OH) deployed just below the surface of the water. Cumulative mortality was tracked by counting dead and live oysters in replicate bags ($n = 2$ bags) bi-weekly (2022) or weekly (2023). Sampling events were conducted at the onset of mortality (0.7–9.4% cumulative mortality) and as mortality was ongoing (20.5–24.8% cumulative mortality). For bacterial isolation, visceral tissue was aseptically dissected from living oysters ($n = 26$ oysters; initial_2022 = 9, ongoing_2022 = 10, initial_2023 = 3, ongoing_2023 = 4) and homogenized in sterile PBS. Homogenates were serially diluted and plated on thiosulfate–citrate–bile salts–sucrose (TCBS) agar for selective isolation of Vibrio species. Vibrios were cultured at ambient seawater temperature measured in the field during each sampling event to reduce temperature-driven culture bias (Table S1).

### Bacterial isolation and DNA extraction

Individual colonies were selected from TCBS plates and purified through three rounds of streak-plating on TCBS agar. A total of 110 clonal isolates were selected for whole-genome sequencing of 1–10 isolates per oyster. Following initial isolation at ambient seawater temperatures, isolates were routinely grown overnight in tryptic soy broth at 26°C with shaking at 200 rpm. DNA was extracted using the KingFisher Flex (Thermo Fisher Scientific) with MagMAX Microbiome Ultra Nucleic Acid Isolation Kit reagents and according to the manufacturer protocols. DNA concentration and purity were assessed using Qubit 1× dsDNA assay kits.

### Genome sequencing and assembly

Whole-genome sequencing was performed by either SeqCenter (2022) or SeqCoast Genomics (2023) using paired end (2 × 150 bp) reads on the Illumina NextSeq 2000. An average of 1.2 ± 0.2 M raw, paired-end reads were produced per sample. Raw reads were trimmed using Trimmomatic v.0.39 (17) and assembled using SPAdes v.4.0.0 with default settings (18). Assembly quality metrics were assessed using QUAST v.5.2.0 (19), BUSCO v.5.7.1 (20), and CheckM v.1.2.3 (21). Assembly quality metrics for each genome are reported in Table S1.

### Genome annotation and analysis

Genomes were annotated using Prokka v.1.14.6 (22) and Barrnap v.0.9 (https://github.com/tseemann/barrnap) with default settings. Virulence factors were identified using abricate v.1.0.1 (https://github.com/tseemann/abricate) with the VFDB database (4 November 2023) (23). Plasmids were detected and characterized using MOB-suite v.3.1.9 (24).

### Taxonomic and phylogenetic analysis

Taxonomic classification for each isolate was determined using GTDB-Tk assignment (25) (Table S1). To determine phylogeny in the context of publicly available oyster-associated strains, the *hsp60* gene was extracted from all annotated genomes using custom Python scripts. Multiple sequence alignment was performed using MAFFT v.7.490 (26) with default parameters and trimmed using trimAl (27). Maximum-likelihood phylogenetic trees were constructed using IQ-TREE v.2.2.2.7 (28) with the best-fit evolutionary model automatically selected using the ModelFinder Plus approach. Bootstrap support and

approximate likelihood-ratio tests (SH-aLRT) were performed with 1,000 replicates each to assess branch support.

To assess the accuracy of both GTDB-Tk taxonomic assignments and *hsp60*-based phylogeny, Panaroo v1.5.2 (29) was used to generate a core genome alignment of all 110 isolates and a maximum-likelihood phylogenetic tree was constructed as described above. All GTDB-Tk taxonomic assignments and *hsp60*-based clade assignments were supported by core genome phylogeny (Fig. S1).

To identify dominant Vibrio groups associated with eastern oysters throughout mortality events, we established conservative criteria for clade designation that required (i) a minimum of four unique isolates per clade, (ii) consisting of isolates from at least four individual oysters, and (iii) clade representation by isolates from both 2022 and 2023.

## Comparative genomic analysis

Orthologous gene clusters were identified using OrthoFinder v2.5.4 (30) with default parameters, including a sequence identity threshold of 70%. Kyoto Encyclopedia of Genes and Genomes (KEGG) modules were annotated and analyzed using Anvi'o v.8 (31). Module completeness was assessed using the "anvi-estimate-metabolism" function with the "—module-completion-threshold 0.5" parameter to identify modules with >50% completion. Differential enrichment between initial and ongoing mortality isolates was evaluated using Fisher's exact test with Benjamini-Hochberg multiple testing correction ($q$-value < 0.05 considered significant). Conserved gene analysis was performed by identifying genes present in 100% of isolates within each clade using a custom Python script that parsed Prokka annotations and OrthoFinder results.

## Statistical analysis

Clade association indices were calculated by comparing observed versus expected co-isolation frequencies from individual oysters assuming random association. Confidence intervals (95%) were calculated using Wilson score intervals. Analysis was based on oysters with >2 isolates sequenced ($n = 23$). Statistical analyses were performed in R v4.4.2.

## RESULTS AND DISCUSSION

### Overview of study design and sampling

To investigate Vibrio community dynamics during oyster mortality events, we conducted a 2-year field study at a commercial aquaculture lease in North Carolina that experiences predictable, severe summer mortality. By focusing on a single study site, we aimed to minimize geographic environmental variation while capturing temporal changes in the Vibrio population as mortality progressed. Over deployment periods between April and May 2022 and 2023, we tracked cumulative oyster mortality and cultured Vibrio bacteria from oyster tissue at the onset of mortality (0.7–9.4% cumulative mortality) and 1 to 2 weeks later as mortality became more severe (20.5–24.8% cumulative mortality). Total cumulative mortality reached 85.4% and 93.8% of deployed oysters in 2022 and 2023, respectively. Throughout the sampling campaigns, we observed only moderate environmental fluctuation between sampling events within a given year (Fig. S2).

Recent pathological investigations of eastern oyster mortality events in North Carolina have identified the digestive system as a primary site of tissue damage and potential bacterial infection (5). Building on this insight, we targeted our microbial sampling to visceral tissues from living oysters at each sampling event. A total of 110 clonal Vibrio strains were isolated from 26 oysters that were alive at the time of sampling, with 1–10 isolates per oyster selected at random for whole-genome sequencing and subsequent analysis.

## *Vibrio* community composition shows consistent succession patterns during mortality events

The 110 genomes sequenced in this study comprised two groups: (i) isolates that were assigned to 20 recognized *Vibrio* or *Photobacterium* species and (ii) additional *Vibrio/Photobacterium* strains not assigned to any previously described species. Among the 20 known species, five were recovered only during the initial low-mortality sampling period, four were detected throughout the study regardless of mortality, and 10 appeared exclusively as mortality was ongoing (Table 1). Phylogenetic analysis based on alignments of core genomes (Fig. S1) and the conserved heat shock protein 60 (*hsp60/groL/cpn60*) gene sequence (Fig. 1) revealed clear temporal patterns in Vibrio community structure throughout mortality progression.

We identified six conserved clades that were consistently associated with eastern oysters (Fig. 2A). The broader Vibrio clade (R2) and Photobacterium clade (R3) were each consistently isolated during both initial and ongoing mortality and are thus referred to as resident populations. The Vibrio resident population consists of strains with no known species assignment but are most closely related to the *V. campbelli*-like strain PID17_43, isolated from nearby Bogue Sound, NC, in 2011 (32). The resident Photobacterium clade consists of *P. damselae*, *P. swingsii*, *P. alginatilyticum,* and *Photobacterium* spp. Only one oyster-associated reference genome, *P. damselae* strain M14-00202 isolated from a Pacific oyster (33), clustered within either of these groups. The existence of various resident Vibrio populations has been documented in marine invertebrates, suggesting these clades may represent core microbiome members with important ecological functions (34, 35).

In contrast to resident groups that maintained their presence throughout mortality, the remaining identified clades were consistently isolated during only one disease stage. Clade E1 consisted exclusively of *V. mediterranei*-clade isolates collected during initial sampling events each year. The E1 clade includes closely related strains identified as *V.*

**TABLE 1** *Vibrio* species isolated in this study[a]

| Species | Initial | Ongoing | Total |
|---|---|---|---|
| *Photobacterium alginatilyticum* | – | 1 | 1 |
| *Photobacterium damselae* | 8 | 12 | 20 |
| *Photobacterium* spp. | – | 2 | 2 |
| *Photobacterium swingsii* | 2 | 1 | 3 |
| *Vibrio aestuarianus* | 3 | – | 3 |
| *Vibrio agarivorans* | 1 | 9 | 10 |
| *Vibrio alginolyticus* | 1 | 3 | 4 |
| *Vibrio barjaei* | 3 | – | 3 |
| *Vibrio celticus* | 1 | – | 1 |
| *Vibrio chaetopteri* | – | 1 | 1 |
| *Vibrio chagasii B* | – | 2 | 2 |
| *Vibrio diabolicus* | – | 3 | 3 |
| *Vibrio fortis* | – | 1 | 1 |
| *Vibrio hangzhouensis* | 1 | – | 1 |
| *Vibrio harveyi* | – | 6 | 6 |
| *Vibrio jasicida* | – | 3 | 3 |
| *Vibrio maritimus B* | – | 1 | 1 |
| *Vibrio mediterranei* | 6 | – | 6 |
| *Vibrio owensii* | – | 4 | 4 |
| *Vibrio parahaemolyticus* | – | 1 | 1 |
| *Vibrio rotiferianus* | 1 | 1 | 2 |
| *Vibrio* spp. | 22 | 10 | 32 |
| Total isolates | 49 | 61 | 110 |

[a]Presence or absence of *Vibrio* species according to GTDB-Tk taxonomic assignment. Numbers indicate isolate counts associated with each mortality stage (absence indicated by –).

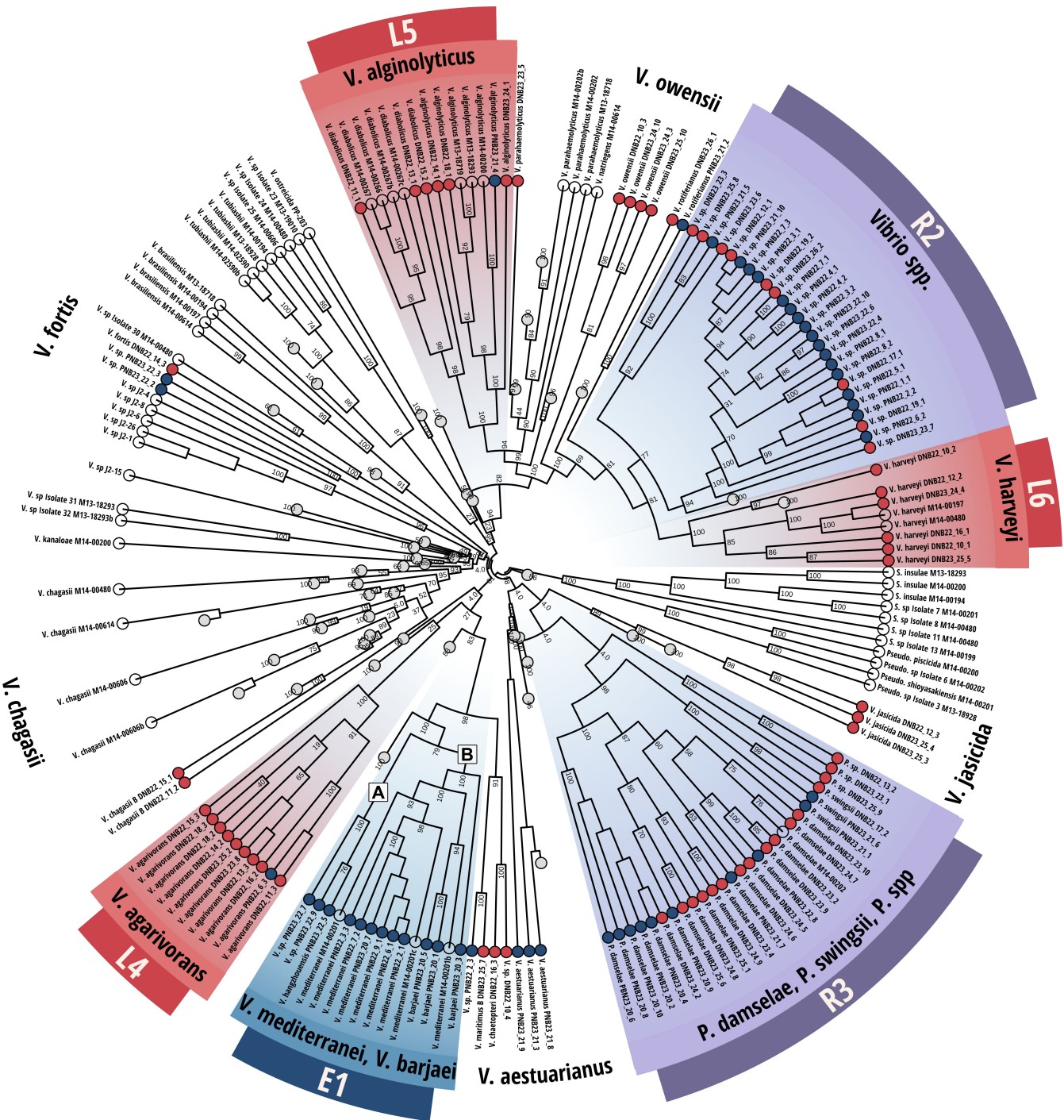

**FIG 1** Phylogeny of Vibrios associated with healthy and diseased eastern oysters. Maximum-likelihood phylogenetic tree based on hsp60 gene sequences (1,680 bp alignment) from 110 Vibrio isolates and 972 reference strains. Tips are colored according to sampling event when strains were isolated (blue = initial; red = ongoing; transparent = reference strains). Conserved clades meeting inclusion criteria are labeled and highlighted (purple = resident clade). Reference Vibrio groups not identified in this data set were retained for phylogenetic context but collapsed for visual clarity. A and B identify *V. mediterranei* (E1) subclades referred to in Table 2. Bootstrap support and approximate likelihood-ratio tests (SH-aLRT) were performed with 1,000 replicates each to assess branch support.

*mediterranei*, *V. barjaei*, *V. hangzhouensis*, or *Vibrio* spp. and is referred to here as the *V. mediterranei* or E1 clade. Clade L6 consisted exclusively of *V. hibrioarveyi* strains isolated as mortality was ongoing. Clades L4 (*V. agarivorans*) and L5 (*V. alginolyticus/V. diabolicus*) were consistently isolated as mortality was ongoing, although each clade is represented

**TABLE 2** Key genes conserved between clades[a]

| Gene annotation | E1A | E1B | R2 | R3 | L4 | L5 | L6 |
|---|---|---|---|---|---|---|---|
| **Secretion systems** | | | | | | | |
| Type IV pilus biogenesis and competence (PilQ) | + | + | + | + | + | + | + |
| Toxin RTX-I translocation ATP-binding protein | ++ | ++ | + | – | + | + | + |
| Type I secretion system (PrsD, PrsE) | ++ | – | – | – | ++ | ++ | ++ |
| Type II secretion system | + | + | + | – | + | + | + |
| Type III secretion system secretin | ++ | ++ | + | – | ++ | ++ | ++ |
| Type III secretion system (YopDGIJKLOQUX) | – | – | – | – | – | + | + |
| Type VI secretion system protein ClpV1 | – | – | – | – | – | ++ | ++ |
| Actin cross-linking toxin VgrG1 | + | – | – | – | – | + | + |
| **Virulence effectors** | | | | | | | |
| Vir factor B; metalloproteases PmbA, TldD | + | + | + | – | + | + | + |
| Protease 3, 4, HtpX, SohB, YhbU | + | + | + | – | + | + | + |
| Protease 2 | – | – | + | – | – | ++ | + |
| Alpha-hemolysin translocation HlyB | + | – | – | + | – | – | – |
| Membrane fusion protein HlyD | ++ | ++ | – | – | + | – | – |
| Virulence protein | + | – | – | – | – | – | + |
| Toxin-antitoxin biofilm protein TabA, toxin CcdB | – | + | – | – | + | – | – |
| Leukotoxin export protein LtxD, tricorn protease | + | – | – | – | – | – | – |
| Toxin YoeB, antitoxin YefM | – | – | – | – | + | – | – |
| Lysophospholipase VolA | – | – | – | – | + | + | + |
| Immunomodulating metalloprotease | – | – | – | – | – | + | – |
| Protein adenylyltransferase VopS | – | – | – | – | – | – | + |
| Microcin C7 self-immunity protein MccF | – | – | – | – | – | – | + |
| **Biofilm and adhesion** | | | | | | | |
| Homoserine efflux protein | ++ | ++ | ++ | – | ++ | ++ | ++ |
| Competence protein ComM | + | + | + | – | + | + | + |
| GlcNAc-binding protein A | ++ | – | + | – | ++ | ++ | ++ |
| Biofilm growth-associated repressor | + | + | – | – | + | – | + |
| Sulfoquinovose isomerase/sulfoquinovosidase | + | + | – | – | – | – | – |
| Invasion protein InvA | – | – | – | – | – | + | + |
| Collagenase; serine proteinase; thermolabile hemolysin | – | – | – | – | – | ++ | ++ |
| Biofilm dispersion protein BdlA | – | – | – | – | – | + | – |
| **Phage-related proteins** | | | | | | | |
| High frequency lysogenization protein HflD | + | + | + | – | + | + | + |
| Phage shock protein ABC; tail-specific protease | + | + | + | – | + | + | + |
| Phage shock protein G | – | + | + | – | + | + | |
| Ferrichrome OM transporter/phage receptor | + | – | – | – | + | + | |
| SPβ prophage-derived aminoglycoside (YokD) | + | + | – | – | – | – | – |
| **Quorum sensing** | | | | | | | |
| AI2 (LuxQ, LuxP) | + | + | + | – | + | + | + |
| CAI-1 (CqsS, synthase) | + | + | – | – | + | + | + |
| Virulence regulator BvgA | + | + | – | – | + | + | ++ |
| Regulators LuxO, SoxR, ToxS | + | + | – | – | – | + | + |
| AI1 sensor kinase/phosphatase LuxN | – | – | – | – | – | + | + |
| Acyl-homoserine-lactone synthase LuxM | – | – | – | – | – | – | + |
| **Iron acquisition** | | | | | | | |
| Iron-uptake and binding (A1, IscA) | + | + | + | – | + | + | + |
| Hemin transport system (HmuTUV) | + | + | – | – | + | + | + |
| Petrobactin transport system (YclNOPQ) | + | – | – | – | + | + | ++ |
| Iron(3+)-hydroxamate transport system (FhuBCD) | + | – | – | – | – | + | + |
| Aerobactin synthase | + | – | – | – | – | – | – |
| Vibriobactin utilization protein ViuB | – | + | – | – | – | – | – |

*(Continued on next page)*

**TABLE 2** Key genes conserved between clades[a] (*Continued*)

| Gene annotation | E1A | E1B | R2 | R3 | L4 | L5 | L6 |
|---|---|---|---|---|---|---|---|
| Cell-surface Mod. | | | | | | | |
| LPS biosynthesis and export | + | + | + | − | + | + | + |
| Lipid A biosynthesis myristolytransferase | + | + | − | − | − | + | + |
| LPS core heptosyltransferase RfaQ | − | + | − | − | − | + | + |
| O-antigen biosynthesis glycosyltransferase WbnH | + | − | − | − | − | − | − |
| Misc | | | | | | | |
| Nodulation protein NodD | + | + | − | − | − | + | + |
| Putative L-ascorbate-6-phosphate lactonase UlaG | + | − | − | − | − | − | + |
| Nif-specific regulatory protein | − | − | − | − | + | − | − |

[a]Genes are grouped into broad functional categories and presented as conserved in single copy (+), conserved in multi-copy (++), or not conserved (−) in each clade. Shaded cells indicate conserved presence within a clade. E1A and E1B represent *V. mediterranei* subclades, as annotated in Fig. 1. R2 = resident Vibrios; R3 = resident Photobacteria; L4 = *V. agarivorans*; L5 = *V. alginolyticus/V. diabolicus*; L6 = *V. harveyi*. Gene annotations were parsed from a presence-absence matrix following OrthoFinder analysis.

by a single isolate during initial sampling. The L5 clade includes closely related strains identified as either *V. alginolyticus* or *V. diabolicus* and is referred to here as the *V. alginolyticus* or L5 clade.

To quantify Vibrio community reorganization throughout mortality events, we calculated the relative abundance of each Vibrio clade across sampling events at both the population level (Fig. 2) and in individual oysters (Fig. 3). When sampling events were ordered by cumulative mortality percentage, clear succession patterns emerged in Vibrio community composition (Fig. 2B). Initial communities were consistently dominated by *V. mediterranei* and resident Vibrios. As mortality progressed, *V. mediterranei* was consistently absent, coinciding with an increase in isolation of *V. harveyi*, *V. alginolyticus*, *V. diabolicus*, and *V. agarivorans* (Fig. 2C). This clade-specific succession was consistent between sampling years despite differences in diverse Vibrio taxa.

## Spatial distribution and ecological relationships between Vibrio clades

To investigate potential ecological relationships between Vibrio clades, we analyzed co-isolation patterns within individual oysters. This analysis compared the probability that two clades would be co-isolated from an individual oyster by random chance

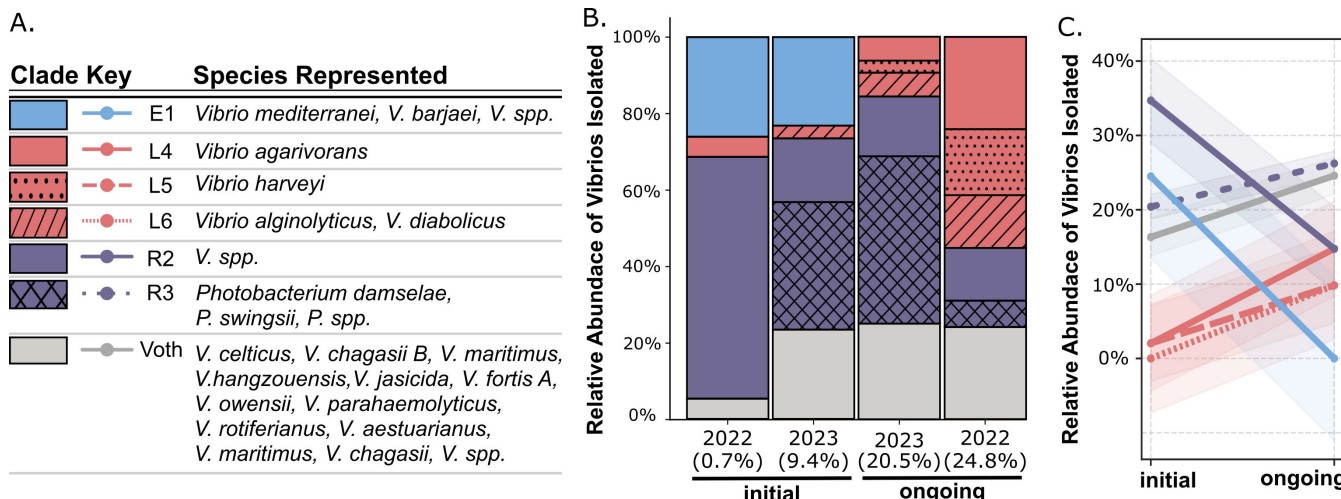

**FIG 2** Consistent community succession occurs as disease progresses. (A) Table indicating clade legend along with all species represented per clade. (B) Relative abundance of Vibrio clades across individual sampling events ordered by percent cumulative mortality (*x*-axis). (C) Line plot showing combined relative abundance of each clade during initial versus ongoing mortality. Negative slope indicates clade decreased in relative abundance; positive slope indicates the clade increased in relative abundance during disease progression.

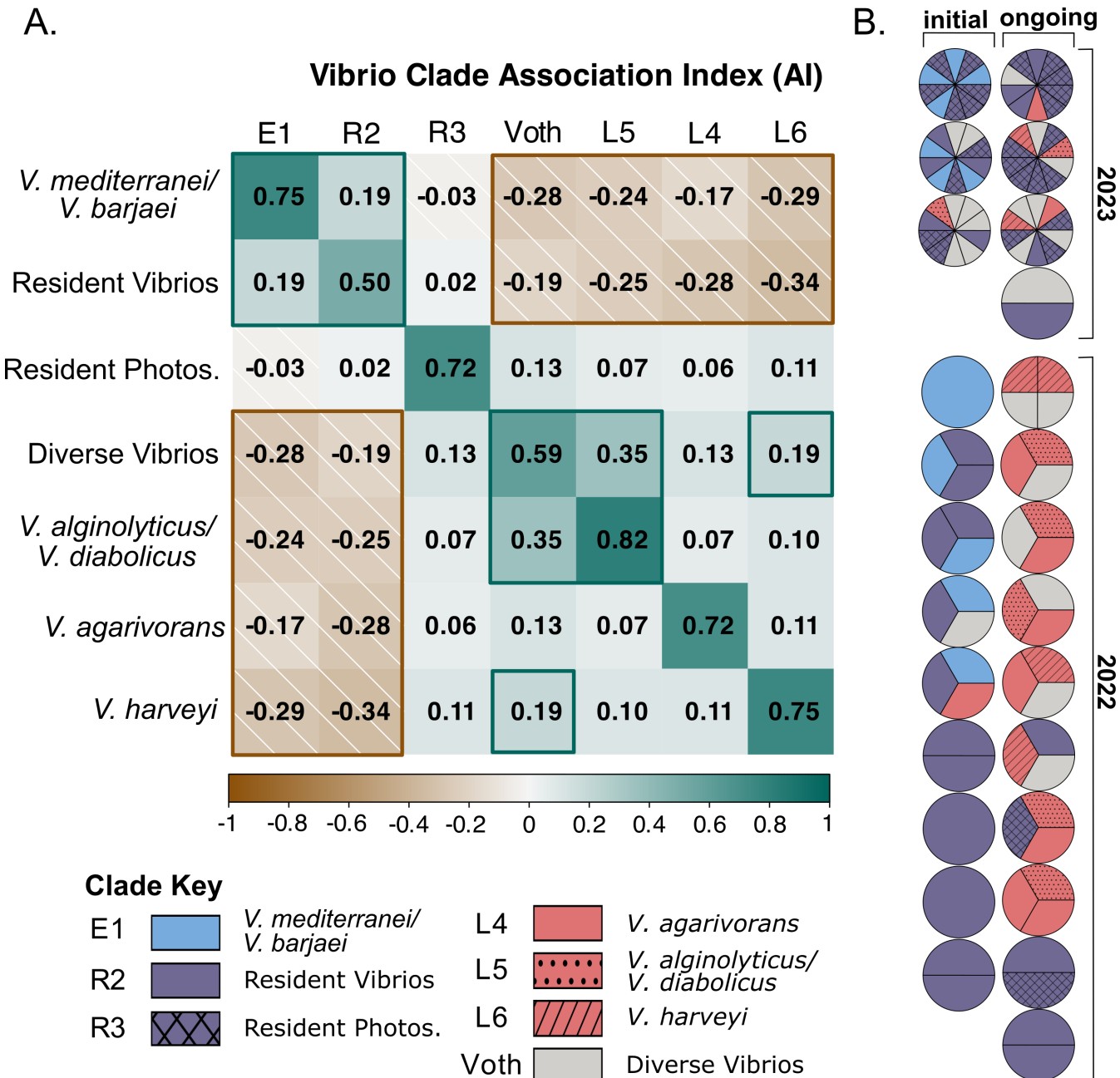

**FIG 3** Vibrio distribution and clade-clade association in individual oysters. (A) Heat map showing association indices between Vibrio clade pairs. Positive (teal) or negative (brown; hatched) values indicates whether clade pair were co-isolated more or less frequently than expected by chance, respectively. Self-association is presented on the diagonal. Analysis included strains from oysters where ≥2 isolates were sequenced. (B) Relative abundance of Vibrio clades isolated from individual oysters, where each animal is represented by a pie plot ($n = 26$).

with the observed co-isolation of clade pairs from oysters with ≥2 isolates sequenced (association index [AI]). All Vibrio clades demonstrated strong self-association (AI = 0.50–0.82), indicating that members of the same clade were isolated together more frequently than expected by chance. Analysis of between-clade relationships revealed that clade interactions largely clustered into two groups of positive and negative associations that closely aligned with initial and ongoing mortality clade designations (Fig. 3B).

*V. mediterranei* and resident Vibrios were positively associated with each other but negatively associated with *V. harveyi*, *V. alginolyticus/V. diabolicus*, *V. agarivorans*, and

diverse Vibrios. Interestingly, both *V. harveyi* and *V. alginolyticus/V. diabolicus* were positively associated with diverse Vibrios but neutrally associated with each other. While resident photobacteria and *V. agarivorans* maintained neutral associations with all other clades, a negative association was observed between *V. agarivorans* and *V. mediterranei/* resident Vibrios.

These spatial distribution patterns reveal a clear ecological partitioning between initial and ongoing mortality-associated Vibrio populations, suggesting that both synergistic and antagonistic relationships may contribute to microbial community organization within oysters. Positive associations between *V. mediterranei* and resident Vibrios specifically in healthy oysters suggest these groups may co-exist commensally or synergistically. In contrast, *V. mediterranei* was never co-isolated with *V. harveyi* or *V. alginolyticus*, despite each individual clade occurring frequently in both sampling years. This complete spatial separation between certain clade pairs indicates that these species may occupy incompatible niches within the oyster microbiome or actively antagonize each other through direct or indirect competitive mechanisms.

The observed transitions between *V. mediterranei*-dominated and *V. harveyi/V. alginolyticus*-dominated communities align with findings from multiple marine systems. Masini et al. documented a marked seasonality in Vibrio composition in seawater, with *V. mediterranei* dominating at temperatures between 15°C and 20°C, while *V. harveyi* and *V. alginolyticus* became dominant at temperatures above 20°C (36). Temperature-dependent host interactions have been reported in coral systems, where *V. mediterranei* and *V. coralliilyticus* co-infection experiments demonstrated enhanced pathogenicity at elevated temperatures (37). The physiological sensitivity of *V. mediterranei* to environmental factors is further evidenced by studies showing that increased salinity can alter its antibiotic resistance profiles (38) and its quorum sensing phenotypes are modulated in response to environmental conditions (39). The consistent nature of these transitions across marine invertebrate hosts suggests conserved ecological interactions that may be driven by environmental factors, host physiology, or interbacterial competition.

## Genomic specialization underlies distinct ecological roles of Vibrio clades

Strong community succession and co-isolation patterns identified specific Vibrio groups associated with initial or ongoing mortality, with this pattern remaining consistent across two years of sampling. To investigate the genomic basis underlying these clade-specific ecological patterns, we first analyzed the distribution of orthologous genes among Vibrio clades. OrthoFinder analysis revealed that the total number of orthologs in each clade ranged from 4,500 to 9,536 (Fig. S3). Resident Photobacterium encoded the greatest number of clade-specific orthologs (2,960), consistent with both the representation of multiple species within this clade and their phylogenetic divergence from the *Vibrio* genus. Notably, we identified 1,280 *V. mediterranei*-specific orthologs that were lost with *V. mediterranei* in the transition from initial to ongoing mortality. Concurrently, 325 *V. harveyi*-specific orthologs were introduced as cumulative mortality increased, along with 126 orthologs specific to *V. harveyi* and *V. alginolyticus*.

To further characterize potential ecological niches of Vibrio clades associated with healthy or diseased oysters, we examined the distribution of key conserved genes between clades (Table 2). The *V. mediterranei* clade (E1) encoded subtle but significant genetic distinctions between two phylogenetically separated subpopulations (E1A and E1B; Fig. 1). Here, we define conserved genes as present in 100% of genomes associated with a given clade and demonstrating >70% sequence identity as calculated by OrthoFinder's default parameters.

Clade-specific gene conservation analysis revealed distinct functional adaptations across Vibrio clades. Most notably, *V. mediterranei* exhibited several unique genomic features related to host colonization and immune evasion. Subclade E1A (*V. hangzhouensis, Vibrio* sp.) uniquely maintained the complete pathway for sulfoquinovose (SQ) utilization, a capacity to metabolize photosynthetically derived compounds (40) that suggests adaptation to algal substrates within the oyster digestive system. Also unique

to *V. mediterranei* was the SPβ prophage-derived aminoglycoside N3′acetyltransferase YokD for antibiotic resistance (41), a type I secretion system, O-antigen biosynthesis glycotransferase WbnH (42, 43), and specialized iron acquisition systems via either aerobactin synthase (E1A) or vibriobactin utilization (E1B).

Conserved genes among clades associated with ongoing mortality were largely involved in host invasion, virulence effectors, and interbacterial competition. The *V. harveyi* clade (L6) encodes a conserved T3SS effector VopS, which targets host cell actin cytoskeleton through AMPylation of Rho-family GTPases, leading to cell rounding and cytoskeletal collapse (44, 45). Both *V. harveyi* and *V. alginolyticus* clades (L5, L6) encode conserved T3SS machinery with YopJ homologs known to block MAPK signaling pathways in host cells (46), as well as multiple metalloproteases, collagenases, toxin/antitoxins, and the surface-anchored lysophospholipase VolA (47). The T3SS has been demonstrated as a crucial virulence mechanism in *V. harveyi* for its role in modulating immune responses in marine invertebrates (48) and could provide mechanistic insight into tissue damage previously observed in moribund eastern oysters (5).

Further analysis of conserved metabolic capacities revealed clade-specific specializations that may influence ecological niches within the host. The *V. agarivorans* clade (L4) was uniquely characterized by a conserved nitrogenase (Nif) cluster, suggesting this clade has the potential to fix nitrogen. The agarolytic capacity of *V. agarivorans*, including conserved pathways for D-galactose catabolism, may provide a selective advantage in degrading complex polysaccharides from red algae cell walls derived from the oyster's planktonic diet.

We identified several conserved genes with potential roles in host–microbe signaling across multiple clades. Notably, nodulation protein D (NodD) homologs were conserved in both *V. mediterranei* and clades associated with ongoing mortality (*V. alginolyticus, V. harveyi*). NodD is classically associated with the initiation of plant-microbe symbioses rhizobia (49, 50). NodD homologs have been documented in *Vibrio vulnificus* (51), suggesting a yet uncharacterized role for NodD in Vibrios. Similarly, UlaG, a manganese-dependent metallo-beta-lactamase involved in L-ascorbate metabolism, was conserved in both *V. mediterranei* (E1) and *V. harveyi* (L6) clades. UlaG belongs to the UlaG-like protein family predominantly found in animal and plant-associated bacteria, where it is thought to confer growth advantages under changing environmental conditions (52, 53). The conservation of these genes among temporally distinct clades hints at a role for these genes in microbial adaptation to the oyster environment.

## Distribution of mobile genetic elements across Vibrio clades

Analysis of mobile genetic elements further differentiated the Vibrio clades examined here. Resident photobacterium strains (R3) showed the highest plasmid diversity, with several strains harboring conjugative plasmids with >97% similarity to pPHDP70 (encoding siderophore biosynthesis) and pHDD1 (encoding hemolytic toxins) (54) (Table S2). Several *V. mediterranei* (E1) strains carry non-mobilizable plasmids that shared between 91 and 93% similarity to the uncharacterized *V. mediterranei* plasmid CP033579. A single *V. alginolyticus* strain encodes a plasmid most similar to pMBL128, identified in a clinical *V. alginolyticus* isolate (55, 56).

Additionally, numerous prophage elements, including SPß-prophage and zonula occludens toxin (Zot), were conserved among *V. mediterranei* strains. Our collection of *V. mediterranei* also harbored strain-specific variations of the TCP-PAI pathogenicity island associated with virulence in larval razor clams (57). CRISPR proteins (Cas2, Cas6, Csx16, Csx10) were uniquely conserved in resident Vibrio and Photobacterium populations, suggesting phage resistance mechanisms that could contribute to their persistent colonization in oysters, even when other clades are displaced (58, 59). These mobile genetic elements could contribute to host–microbe interactions through multiple mechanisms, including superinfection exclusion, quorum-sensing-mediated prophage regulation, and cell-surface modifications affecting adhesion (60).

## Metabolic transitions signal shift from commensalism to potential pathogenicity

Given the distinct genetic repertoires of oyster-associated Vibrios and the temporal separation observed between Vibrio clades, we hypothesized that broader metabolic capabilities may be associated with oysters based on disease state. We conducted KEGG module enrichment analysis to assess functional differences between Vibrios isolated at disease onset versus during ongoing mortality, providing insight into community-level transitions during disease progression.

Our analysis revealed a striking genome-level metabolic transition from biosynthetic to degradative processes as mortality progressed. At the onset of mortality, the Vibrio community showed selective enrichment in arginine biosynthesis pathways proceeding from glutamate via acetylcitrulline (M00845) (Fig. 4). This biosynthetic capacity suggests metabolic independence from host-derived resources. In contrast, the ongoing-mortality community was significantly enriched in hydroxyproline degradation pathways converting trans-4-hydroxy-L-proline to 2-oxoglutarate (M00948), indicating adaptation to increased availability of host-derived proteins as tissue integrity deteriorates.

Nucleic acid metabolism was distinctly associated with initial communities, with enrichment in both purine (M00958, M00959, M00546) and pyrimidine (M00046) degradation pathways. These communities were also enriched in aromatic compound degradation modules, including catechol meta-cleavage (M00569) and benzoate degradation (M00540). This capacity to metabolize complex carbon sources suggests adaptation to the diverse substrates available in the healthy oyster digestive environment and potential contribution to nitrogen cycling within the host.

Enrichment in hydroxyproline catabolism during ongoing mortality is particularly significant as it provides direct genomic evidence for collagen degradation. Hydroxyproline is a modified amino acid highly abundant in collagen, serving as a specific marker for collagen breakdown (61). This finding connects genomic data with previous

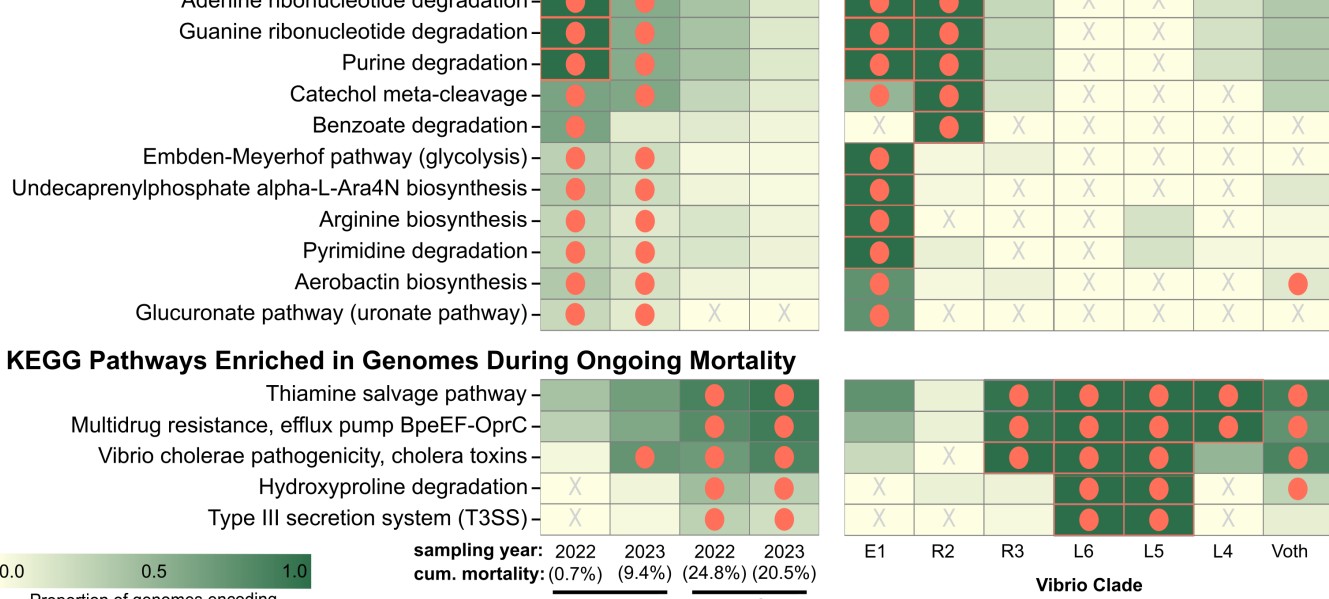

**FIG 4** Differential enrichment of metabolic pathways in communities associated with initial and ongoing mortality. Heat map showing the proportion of genomes encoding KEGG modules in each Vibrio clade. Data are separated into complete pathways enriched during initial (top) versus ongoing (bottom) mortality. For comparison, differential enrichment analysis was repeated according to clade and presented in the corresponding panel. Color intensity reflects the proportion of genomes in each group that encode complete pathway modules (0–1). Orange circles indicate significantly enriched modules; adjusted *q*-value <0.05.

pathological observations of tissue damage in moribund eastern oysters (5). Collagen-derived peptides are known to provide protective functions in marine invertebrates, including free radical scavenging and anti-inflammatory activity (62, 63), suggesting that collagen degradation may not only provide nutritional benefits to opportunistic bacteria, but also the potential to compromise host defenses.

The ongoing mortality Vibrio community also exhibited significant enrichment in pathogenicity-associated modules, including multidrug resistance (M00698), type III secretion systems (T3SS; M00542), and cholera toxin production (M00850). This expansion of virulence-associated pathways coincided with enrichment in thiamine salvage (M00899), a stress response mechanism activated under deteriorating environmental conditions that has been linked to enhanced pathogenicity in other host-pathogen systems (64).

This metabolic transition suggests a fundamental shift in how Vibrios interact with their oyster host during mortality events. Our data indicates that initial communities display genomic signatures of stable colonization without host degradation, suggesting primarily commensal or symbiotic interactions. In contrast, ongoing-mortality communities encode genomic adaptations that would enable tissue degradation and active host manipulation through diverse virulence factors. While further experimentation is required to confirm the *in situ* expression of these pathways, this genomic evidence provides a mechanistic framework for understanding how Vibrio-oyster interactions evolve from commensal to opportunistic or pathogenic as mortality progresses.

## Vibrio community succession and disease progression

The consistent dominance of *V. mediterranei* in initial bacterial communities presents an intriguing contrast with its documented pathogenicity in corals and clams (15, 65, 66). Temperature-mediated pathogenicity has been established in *V. mediterranei*'s interactions with the pen shell clam, *Pinna nobilis* (15). Additionally, Rubio-Portillo et al. demonstrated enhanced virulence toward the coral *Oculina patagonia* in co-culture with other Vibrio species (65). However, *V. mediterranei* has also been associated with healthy Pacific oysters and scallops (33, 67), suggesting environmental context and polymicrobial interactions may play a fundamental role in determining the ecological outcomes of host–microbe relationships.

In this study, we observed a transition from *V. mediterranei*-dominated communities to assemblages enriched in *V. harveyi*, *V. alginolyticus*, *V. diabolicus*, and *V. agarivorans*, which represented a critical threshold in disease progression occurring in the 1–2 weeks prior to the onset of observed mortality. Genomic evidence suggests this transition reflects a fundamental shift from commensal metabolism (arginine biosynthesis, nutrient cycling, aromatics degradation) to degradative or pathogenic activity (T3SS, virulence effectors, hydroxyproline degradation). However, experimental evidence is required to determine if or how strains isolated in this study ultimately contribute to mortality events. This parallels patterns observed in other host–microbe systems where environmental stress triggers the collapse of beneficial relationships (68). Multiple interconnected mechanisms likely drive this community reorganization. Polymicrobial synergy may represent a key factor, as demonstrated in Pacific oyster systems where *V. harveyi* simultaneously dampens host immune responses while producing siderophores that benefit co-existing Vibrios (69). The strong negative associations observed between initial and ongoing-mortality clades (Fig. 3) suggest active competitive exclusion through resource limitation, direct antagonism, or differential responses to host physiological states. The consistency of this rapid transition across both sampling years indicates it represents a reproducible critical phase that could be utilized as a predictive indicator for severe mortality.

## Implications for aquaculture management

These findings suggest that monitoring commensal microbial populations and community transitions could provide earlier warning of impending mortality events

than traditional pathogen monitoring alone. The rapid timeline of community reorganization provides a practical window for potential intervention strategies while emphasizing the need for regular microbial monitoring. For instance, digital PCR targeting *V. mediterranei*-specific markers or microbiome profiling using the *hsp60* gene (7, 70) could enable cost-effective early warning in commercial settings (71). This approach aligns with emerging ecological disease management strategies and recent research demonstrating that maintaining rich pre-existing microbiomes can limit pathogenic Vibrio colonization (72).

## Limitations and future directions

Several limitations of this study should be acknowledged. Genomic content does not reflect *in situ* gene expression; transcriptomic and metabolomic approaches will be essential to confirm which pathways are actively expressed during different mortality stages. Culture-based methods often underestimate Vibrio diversity, as some species may be difficult to cultivate under laboratory conditions (73). Finally, sampling from a single aquaculture farm limits our ability to generalize findings across broader geographic regions.

Future research should focus on experimental validation of the ecological niches proposed here. Controlled microbial challenge experiments could determine whether clades identified here actively contribute to disease progression or simply serve as an indicator of oyster condition. Similarly, experiments testing the combined effects of temperature, salinity, and microbial community structure will help identify the environmental triggers that initiate community reorganization.

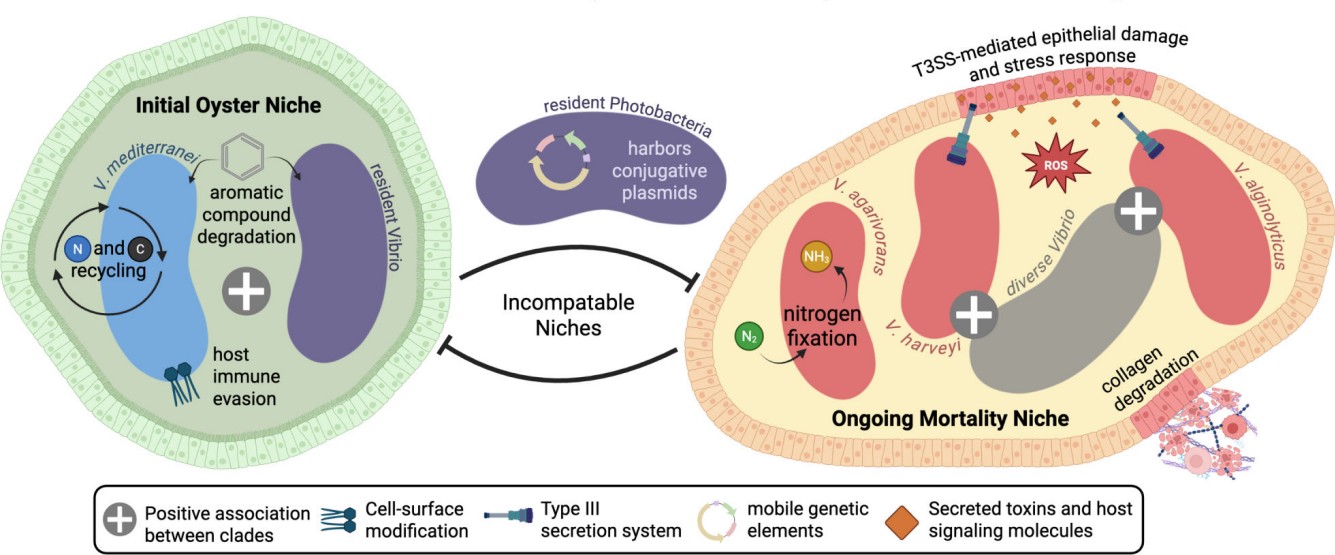

**FIG 5** Proposed model of Vibrio niches and community transitions associated with oyster mortality. Summary highlighting the key species and niche functions associated with initial and ongoing mortality in eastern oysters. Community reorganization patterns are indicated, along with predicted changes in the host environment that may drive or respond to microbial community reorganization.

## Conclusions

This study identifies genomic evidence for consistent Vibrio community reorganization during oyster mortality events, revealing repeated succession from *V. mediterranei*-dominated communities to communities enriched in *V. harveyi*, *V. alginolyticus*, *V. diabolicus*, and *V. agarivorans* (Fig. 5). This transition reflects a fundamental shift from biosynthetic, commensal metabolism to degradative, potentially pathogenic functions. The conservation of this succession pattern across multiple years suggests these dynamics represent fundamental ecological principles governing host–microbe interactions during disease progression. By integrating microbial community monitoring with traditional management approaches, this research provides a foundation for developing early warning systems and intervention strategies to mitigate mortality events in shellfish aquaculture.

### ACKNOWLEDGMENTS

We gratefully acknowledge the UNCW Shellfish Research Hatchery and Virginia Institute of Marine Sciences for providing the oysters used in this study and our partner farmers for access to their commercial lease for field deployments. We thank Colin Eimers, Tami Bennett, Jonathan Lucas, and Carly Dinga for their assistance with field sampling and bacterial isolation.

This work was partially supported through funding appropriated through the North Carolina General Assembly under NCGS 11-173.1 and approved through the North Carolina Marine Fisheries Commission and the Funding Committee for the North Carolina Commercial Fishing Resource Fund. This work was also partially supported through the North Carolina Collaboratory and the U.S. Department of Agriculture's National Institute of Food and Agriculture A1712 Programs: Rapid Response to Extreme Weather Events Across Food and Agricultural Systems, Proposal Systems, Proposal Number 2024-68016-42233.

S.S. and M.C. conceptualized this study; M.C. and T.C. performed field collections; S.S. performed data curation, computational analysis, and writing; T.B.-H. and R.T.N. acquired funding and resources for this study.

### AUTHOR AFFILIATIONS

[1]Department of Earth, Marine and Environmental Sciences, Institute of Marine Sciences, University of North Carolina, Morehead City, North Carolina, USA

[2]Department of Clinical Sciences, College of Veterinary Medicine, North Carolina State University, Morehead City, North Carolina, USA

### AUTHOR ORCIDs

Steph Smith http://orcid.org/0000-0002-3709-4464
Mark Ciesielski http://orcid.org/0000-0002-8996-2323
Tal Ben-Horin http://orcid.org/0000-0001-5123-2643

### FUNDING

| Funder | Grant(s) | Author(s) |
| --- | --- | --- |
| U.S. Department of Agriculture | 2024-68016-42233 | Rachel T. Noble |

### AUTHOR CONTRIBUTIONS

Steph Smith, Conceptualization, Data curation, Formal analysis, Investigation, Methodology, Visualization, Writing – original draft, Writing – review and editing | Mark Ciesielski, Conceptualization, Data curation, Investigation, Writing – review and editing | Thomas Clerkin, Investigation, Writing – review and editing | Tal Ben-Horin, Funding acquisition, Writing – review and editing | Rachel T. Noble, Funding acquisition, Writing – review and editing

## DATA AVAILABILITY

The genome assemblies generated and analyzed in this study are available in the GenBank repository under Project Accession PRJNA1250144.

## ADDITIONAL FILES

The following material is available online.

### Supplemental Material

**Supplemental Material (mSystems01078-25-s0001.docx).** Figures S1 to S3 and Table S2.
**Table S1 (mSystems01078-25-s0002.xlsx).** Strain data.

### Open Peer Review

**PEER REVIEW HISTORY (review-history.pdf).** An accounting of the reviewer comments and feedback.

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
