## [Reviewer comments · mSystems]

Farmed oyster mortality follows consistent *Vibrio* community reorganization

Steph Smith, Mark Ciesielski, Thomas Clerkin, Tal Ben-Horin, and Rachel Noble

Corresponding Author(s): Steph Smith, The University of North Carolina at Chapel Hill

Review Timeline:

Submission Date:

July 18, 2025

Accepted:

September 14, 2025

Editor: Suzanne Ishaq

Reviewer(s): Disclosure of reviewer identity is with reference to reviewer comments included in decision letter(s). The following individuals involved in review of your submission have agreed to reveal their identity: Jeffrey W Turner (Reviewer #3)

Transaction Report:

DOI: <https://doi.org/10.1128/msystems.01078-25>

Re: mSystems01078-25 (**Farmed oyster mortality follows consistent Vibrio community reorganization**)

Dear Dr. Steph Smith:

Your manuscript has been accepted, and I am forwarding it to the ASM production staff for publication. Your paper will first be checked to make sure all elements meet the technical requirements. ASM staff will contact you if anything needs to be revised before copyediting and production can begin. Otherwise, you will be notified when your proofs are ready to be viewed.

Sincerely,
Suzanne Ishaq
Editor
mSystems

Reviewer #3 (Comments for the Author):

This article details a fascinating microbial succession during oyster mortality. This research is of great importance and interest and the illustrations are engaging. The authors addressed the reviewer comments completely and the revision is much improved. I recommend accepting the article with no further revisions.